# GUARD: Guiding Unbiased Alignment through Reward Debiasing

## Abstract

Reward misspecification in RLHF threatens the reliability of large language models by amplifying spurious correlations and producing unstable or unsafe behavior Christiano et al. [2017], Skalse et al. [2022], Gao et al. [2023]. Expert-defined harm categories provide a stable signal for post-training evaluation Mitchell et al. [2019], but reward models often encode categorical biases that undermine trustworthiness. We address this challenge through an information-theoretic reliability objective: minimizing mutual information Belghazi et al. [2018] between reward scores and sensitive categories. Our approach enforces invariance via adversarial training Edwards and Storkey [2016], Zhao et al. [2018] while integrating curiosity-driven intrinsic rewards Pathak et al. [2017] into PPO Schulman et al. [2017] to preserve diversity. Framing debiasing as a minimax game yields reward models that are both robust and verifiably category-independent. Empirically, our Fair-RM achieves near-neutral bias on CrowS-Pairs Nangia et al. [2020] and StereoSet Nadeem et al. [2020], reduces post-PPO disparity on HH-RLHF, and scales to 19-category fairness in PKU-SafeRLHF Ji et al. [2024]. These results demonstrate improved calibration and stability under distribution shift, establishing our method as a practical reliability control for safety-critical RLHF deployment.

## 1 Introduction

Reinforcement Learning from Human Feedback (RLHF) has become essential for aligning large language models with human intent Christiano et al. [2017], Ouyang et al. [2022], yet reward misspecification poses significant risks for reliability in safety-critical applications Amodei et al. [2016], Pan et al. [2022]. When reward models inherit biases from pretraining or exploit spurious correlations Skalse et al. [2022], downstream policies can display unstable or unsafe behaviors across demographic groups or safety categories—a major barrier to deployment in domains such as healthcare, finance, and criminal justice. These failures undermine not only fairness but also calibration, robustness, and the broader trustworthiness of RLHF systems.

Existing approaches to mitigating bias typically rely on penalty-based regularization Shen et al. [2023], Dai et al. [2023] that augments the training loss, or resource reallocation across groups Ouyang et al. [2025] and ensemble-based multi-objective methods Zhou et al. [2024]. While such techniques reduce observed disparities, they lack theoretical guarantees of reliability, often collapse under distribution shift, and may sacrifice response diversity. As a result, these strategies leave open important failure modes—including reward hacking and instability—that limit confidence in their use for safety-critical AI deployment.

Our key insight is that reliability can be formalized as statistical independence between reward outputs and sensitive categories Belghazi et al. [2018], Zhao et al. [2018]. We implement this by introducing an adversarial minimax game Edwards and Storkey [2016] that enforces invariance in the reward model while preserving preference learning performance. To counteract the reduction in generative diversity that such constraints can impose, we further integrate a curiosity-driven intrinsic reward

Submitted to the NeurIPS 2025 Workshop on Reliable Machine Learning from Unreliable Data. Do not distribute.

during PPO training Pathak et al. [2017], Schulman et al. [2017]. Together, these components form a principled and scalable framework that embeds reliability requirements directly into the reward modeling stage, enabling verifiable improvements in calibration, robustness, and fairness across diverse categories.

## 2 Related Work

**Reward Misspecification and Reliability in RLHF.** Prior work has identified reward misspecification as a fundamental threat to RLHF reliability, including reward hacking and over-optimization Skalse et al. [2022], Gao et al. [2023]. Existing mitigation strategies—penalty-based regularization Shen et al. [2023], Dai et al. [2023], resource reallocation Ouyang et al. [2025], and multi-objective methods Zhou et al. [2024], Wu et al. [2023]—lack theoretical guarantees and often collapse under distribution shift. Our work formalizes reliability as statistical independence with verifiable adversarial constraints.

**Information-Theoretic Fairness and Adversarial Training.** Mutual information has been used to enforce fairness through adversarial training that minimizes dependence on sensitive attributes Edwards and Storkey [2016], Zhao et al. [2018], Belghazi et al. [2018]. Parallel work explores adversarial and self-play approaches to better represent heterogeneous preferences and bypass reward models Cheng et al. [2024], Wu et al. [2024], Chen et al. [2024], Bukharin et al. [2025], Wang et al. [2025, 2024]. We combine adversarial debiasing with curiosity-driven rewards Pathak et al. [2017] to enforce category independence while preserving diversity during PPO training.

## 3 Problem Setup and Method

**Reward Modeling in RLHF.** An RLHF reward model (RM) assigns a scalar score $r_\theta(x, y)$ to a prompt–response pair and is trained from human pairwise preferences Christiano et al. [2017], Ouyang et al. [2022]. We use the Bradley–Terry formulation Bradley and Terry [1952]

$$P(y_A \succ y_B) = \sigma\big(r_\theta(x, y_A) - r_\theta(x, y_B)\big),$$

with training objective (averaged over pairs)

$$L_{\mathrm{BT}}(\theta) = -\log \sigma\big(r_\theta(x, y_A) - r_\theta(x, y_B)\big),$$

so minimizing $L_{\mathrm{BT}}$ drives $r_\theta(x, y_A) > r_\theta(x, y_B)$ when $y_A$ is preferred. The BT objective represents an MLE of the preference dataset onto the space of scalar-valued reward models Swamy et al. [2025].

**Reliability Constraint via Mutual Information.** Following Ouyang et al. [2025], we treat reliability of an RM across categories $c \in C$ (e.g., helpfulness/harmlessness or broader safety tags) as *invariance* of the reward scale with respect to these categories (see Appx. A.1 for how non-invariant RMs can induce undesirable downstream behavior). Formally, we target identical reward distributions $r_\theta(x, y \mid c)$ for all $c$, i.e.,

$$I\big(r_\theta(x, y); c\big) = 0,$$

zero mutual information between reward and category Belghazi et al. [2018], Zhao et al. [2018]. Directly minimizing this dependence is intractable, so we adopt an adversarial surrogate: a classifier $q_\phi(c \mid r)$ attempts to predict $c$ from rewards. This casts reliable (category-invariant) reward learning as a minimax game between the reward model and a discriminator solved via no-regret dynamics; our analysis (Appendix A.3) shows that such training drives the empirical MI toward zero.

**Adversarial Implementation.** We impose the constraint during RM training on preference pairs, where each comparison $(x, y_A, y_B)$ carries a category label. We optimize $L_{\mathrm{BT}}$ for preference prediction while training an adversary $q_\phi$ on scored examples $(x, y)$; a lightweight MLP consumes scalar rewards $r_\theta(x, y_A)$ and $r_\theta(x, y_B)$ to predict $c$. In practice, the adversarial weight $\lambda_{\mathrm{adv}}$ trades off invariance against stability and fit. To preserve output diversity while enforcing invariance, we add a small intrinsic reward via Random Network Distillation (RND) Pathak et al. [2017], Burda et al. [2019] during PPO, following recent introductions of intrinsic reward into RLHF Sun et al. [2025].

## 4 Experiments and Results

We evaluate our framework on a binary Helpful/Harmless (HH-RLHF) task Bai et al. [2022] and a 19-class safety classification task Ji et al. [2024]. We fine-tune TinyLlama-1.1B TinyLlama Team [2024] policies with PPO Schulman et al. [2017], Hugging Face [2023], comparing a baseline reward model against our Fair and Fair+Curiosity variants. Full training and evaluation details are provided in Appendix A.4–B.

**Reward Distribution Analysis.** In our main experiment, we compare reward model scores across Helpful versus Harmless completions. The baseline RM exhibits a systematic skew, consistently inflating Helpful rewards. This distortion allows a weak completion from one category (e.g., unhelpful) to outrank a strong completion from another (e.g., harmless), violating the assumption of a shared reward scale.

Our fairness-constrained model with $\lambda_{adv} = 0.2$ produces a substantially more balanced distribution (Figures 5, 6). The KS distance decreases from 0.43 to 0.10 ($p < 0.001$) and the Wasserstein-1 distance from 13.38 to 0.53 ($p < 0.001$), reflecting a statistically significant reduction in categorical bias. This enforces comparability of rewards across behavior types, yielding more reliable evaluations; a post-hoc predictability test (Appx. A.7) confirms that category membership is nearly unrecoverable from the debiased rewards.

Hyperparameter settings are given in Appendix A.6, with MI estimator details in Section A.8.

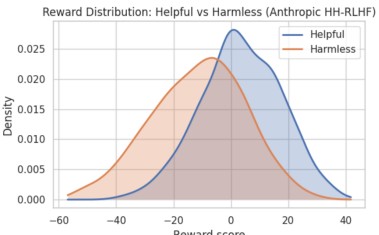
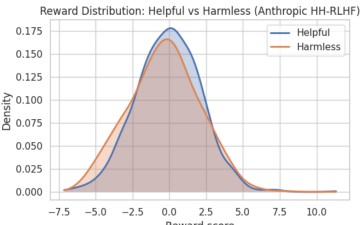

Figure 1: Reward distribution before applying fairness constraint

Figure 2: Reward distribution after applying fairness constraint

## 4.1 Post-PPO Fairness

After PPO fine-tuning on HH-RLHF, we evaluate all policies on 100 Helpful and 100 Harmless prompts, scoring with an HH-RLHF-trained safety RM Bai et al. [2022]. The baseline policy exhibits a parity gap of 0.4814, reduced to 0.4001 ($-16.9\%$) under the fairness constraint and 0.4126 ($-14.3\%$) with Fair+Curiosity. Curiosity slightly widens the gap relative to fairness alone but still markedly improves over baseline while recovering most variance and response diversity. See Sec. 4.1 and Appx. B.1 for additional discussion.

| Policy | Parity Gap | Relative Drop |
|---|---|---|
| Baseline | 0.4814 | – |
| Fair | 0.4001 | $-16.9\%$ |
| Fair + Curiosity | 0.4126 | $-14.3\%$ |

Table 1: Parity gap between Helpful and Harmless mean rewards on HH-RLHF prompts post-PPO.

**Diversity.** We measure *semantic diversity* via average pairwise cosine distance of `all-mpnet-base-v2` embeddings Reimers and Gurevych [2019], Song et al. [2020]; details are given in Appx. B.2. Fairness alone reduces diversity from 0.9638 to 0.9584 ($p < 0.001$), while adding curiosity restores it to 0.9616 ($p = 0.002$), nearly recovering baseline levels. This indicates that curiosity mitigates the diversity loss induced by fairness regularization. Results are reported from early-stage PPO training; longer runs may amplify these effects, which we leave to future work.

## 4.2 Generalization to Unseen Biases

**Setup** We train two HH-RLHF reward models Bai et al. [2022]: a baseline ($\lambda_{adv} = 0$, Bradley–Terry) and a fairness-constrained model ($\lambda_{adv} = 0.2$, MI penalty). Bias is assessed on CrowS-Pairs Nangia et al. [2020] and StereoSet Nadeem et al. [2020] as the proportion of stereotypical predictions (neutral = 50%).

**Results** Table 2 shows that introducing the MI constraint shifts bias rates toward neutrality compared to the baseline RM, with statistically significant improvements (CrowS-Pairs: McNemar $p<0.001$; StereoSet: $p<0.01$). Notably, the fairness objective is trained without access to CrowS-Pairs or

StereoSet, yet reduces stereotype bias across domains. This demonstrates generalization beyond training categories and highlights a scalable path to mitigating unseen RLHF biases.

| Model | CrowS-Pairs Bias (%) | StereoSet Bias (%) |
|---|---|---|
| Baseline RM | $42.84\% \pm 1.27\%$ | $46.58\% \pm 1.09\%$ |
| Fair RM | $51.46\% \pm 1.29\%$ | $49.95\% \pm 1.09\%$ |

Table 2: Generalization results. Bias rates measure preference for stereotypical sentences (50% = neutral). Values show mean $\pm$ standard error.

### 4.3 Fairness Across Multiple Harm Categories

**Setup** We train two Llama-3.2-1B reward models on the 19-category PKU-SafeRLHF dataset Ji et al. [2024]: a *Baseline* ($\lambda_{\text{adv}} = 0$) and a *Fair* model with an MI adversary ($\lambda_{\text{adv}} = 0.2$). While the baseline displays large reward disparities across harm categories, the fairness-constrained RM produces dis-

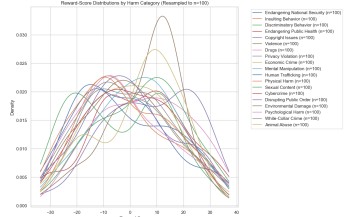
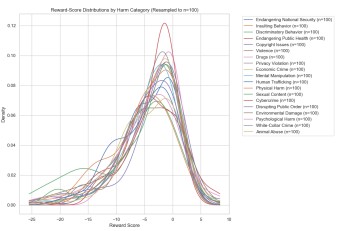

Figure 3: Before fairness.      Figure 4: After fairness.

tributions that are far more uniform. Crucially, the distributions do not collapse; the RM preserves its Bradley–Terry predictive performance, showing that a single model can be made fair across many categories simultaneously—scaling fairness beyond binary setups.

### 4.4 Ablation: Adversarial Weight

**Setup** We analyze the effect of the adversarial weight $\lambda_{\text{adv}}$ on our MI objective by sweeping this parameter (full results in Appx. A.9). For each setting, we report both mutual information (MI) and Bradley–Terry (BT) loss. Table 3 shows a steep drop in MI as $\lambda_{\text{adv}}$ increases, alongside improvements in BT loss. This suggests that the fairness constraint doubles as a regularizer, enhancing preference learning while suppressing categorical dependence.

| $\lambda_{\text{adv}}$ | BT loss | MI |
|---|---|---|
| 0.0 | 2.8712 | 0.2282 |
| 0.2 | 2.2307 | 0.0163 |
| 0.8 | 1.1879 | 0.0073 |
| 1.5 | 0.7432 | 0.0136 |

Table 3: Representative $\lambda_{\text{adv}}$ settings; full sweep in Appx. A.9.

## 5 Conclusion

We introduce an adversarial MI constraint that reduces bias in reward models while keeping alignment with human preferences intact. Across tasks like CrowS-Pairs, StereoSet, and SafeRLHF's 19 categories, our method improves fairness without sacrificing performance. By pairing this with an intrinsic reward in PPO, we position fairness as a built-in reliability goal rather than an add-on. This provides a scalable path toward preference-aligned reward models that are consistent and trustworthy. Looking ahead, we plan to test larger models and study how fairness interacts with emergent behaviors such as reward hacking.

## 6 Ethics and Limitations

Our adversarial training method is motivated by zero-information strategies, but practical noisiness makes it hard to tune Edwards and Storkey [2016], Belghazi et al. [2018]. Its effectiveness depends on well-defined, discrete categories, suggesting future work should extend to non-discrete attributes Mitchell et al. [2019], Bolukbasi et al. [2016]. The approach also increases time and memory costs Ouyang et al. [2022], requiring larger batch sizes for distribution-level statistics, and our experiments remain limited in characterizing the reward hacking dynamics introduced by this constraint.

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

# A Appendix

## A.1 Why enforce fairness on Reward Models?

In this section, we offer an intuitive thought experiment on why fairness defined as categorical independence of the reward model distribution mitigates undesired reward hacking scenarios in PPO. Consider the example given in the main text 4 and suppose $y_{i,c}, y_{i,r}$ are chosen and rejected samples from the $i$th datapoint in our preference dataset respectively. We observe cases where $\exists i, j$ such that $y_{i,c} > y_{i,r} > y_{j,c} > y_{j,r}$. That is, because datapoint $i$ and datapoint $j$ are independent of one another, we can have a *good* model in the Bradley-Terry definition prioritize chosen over rejected within the pair, but then across pairs end up rewarding a rejected sample of one pair over the chosen sample of another. In practice, we notice a systemic shift towards higher rewards for $i \in D_{helpful}$ (the subset of preference exemplars portraying helpful behaviors) over $j \in D_{harmless}$ (the subset of preference exemplars portraying harmless behaviors). Then, for cases where $y_{i,c} > y_{i,r} > y_{j,c}$, we will observe behavior in the post-trained LM where it prioritizes both helpful and unhelpful behavior over harmless behavior given a potentially harmful prompt.

## A.2 Theoretical Justification

We ground our approach in adversarial training theory, considering a reward model $r_\theta : \mathcal{X} \to \mathbb{R}$ and a discriminator $q_\phi(c \mid \cdot)$ Edwards and Storkey [2016].

**Setting.** We observe i.i.d. triples $(X_t^+, X_t^-, C_t)$ with labels $Y_t \in \{0, 1\}$ indicating whether $X_t^+$ is preferred to $X_t^-$ from some unknown preference distribution. Let $R_\theta = r_\theta(X)$. The (population) Bradley–Terry loss is

$$\mathcal{L}_{\text{BT}}(\theta) = \mathbb{E}\big[-\log \sigma\big(r_\theta(X^+) - r_\theta(X^-)\big)\big]. \tag{1}$$

Our discriminator $q_\phi(c \mid \cdot)$ tries to infer $C$ from rewards. We thus have the zero-sum game

$$\min_\theta \max_\phi \ \mathcal{J}(\theta, \phi) = \mathcal{L}_{\text{BT}}(\theta) + \lambda \mathbb{E}\big[\log q_\phi(C \mid R_\theta)\big]. \tag{2}$$

where our target is independence: $R_\theta \perp C$ (i.e., $I_\theta(C; R_\theta) = 0$).

Our main theoretical result connects the adversarial training scheme to our original fairness objective:

**Theorem 1** (No-regret reaches mutual information target). *Assume Lemma 1, feasible invariance* (7), *and no-regret play with* $\text{Reg}_G(T), \text{Reg}_D(T) = o(T)$. *Then*

$$\frac{1}{T} \sum_{t=1}^{T} I_{\theta_t}(C; R_{\theta_t}) \leq \frac{\text{Reg}_G(T) + \text{Reg}_D(T)}{\lambda T} \xrightarrow[T \to \infty]{} 0. \tag{3}$$

## A.3 Proof of Theoretical Results

In this section we provide a proof for our main convergence theorem, starting with supporting lemmas to demonstrate the equivalence of our adversarial game to mutual information minimization.

**Lemma 1** (Best response is a mutual-information penalty). *If we take a fixed $\theta$,*

$$\sup_\phi \mathbb{E}\big[\log q_\phi(C \mid R_\theta)\big] = \mathbb{E}\big[\log p_\theta(C \mid R_\theta)\big] = -H_\theta(C \mid R_\theta).$$

*This implies that the inner game's value is nothing more than $-H_\theta(C \mid R_\theta)$, the negative conditional entropy of categories given the reward model distribution (for a slight abuse of notation), and so the reward model's objective becomes*

$$\overline{\mathcal{J}}(\theta) := \sup_\phi \mathcal{J}(\theta, \phi) = \mathcal{L}_{\text{BT}}(\theta) + \lambda I_\theta(C; R_\theta). \tag{4}$$

*We drop the additive constant $-\lambda H(C)$ since it does not depend on $\theta$.*

*Moreover, any best-response discriminator satisfies $q_{\phi^\star}(\cdot \mid r) = p_\theta(\cdot \mid r)$ a.s.*

We turn to the literature of no-regret algorithms as solvers for two-player zero-sum (2p0s) games to show the convergence of this adversarial training procedure, defining the regret for the reward model and discriminator respectively.

**Repeated play and regrets.** At round $t = 1, \ldots, T$, the reward model chooses $\theta_t$, the discriminator chooses $\phi_t$, and both observe payoff $\mathcal{J}(\theta_t, \phi_t)$. Define external regrets

$$\mathrm{Reg}_G(T) := \sum_{t=1}^{T} \mathcal{J}(\theta_t, \phi_t) - \min_{\theta} \sum_{t=1}^{T} \mathcal{J}(\theta, \phi_t), \qquad \mathrm{Reg}_D(T) := \max_{\phi} \sum_{t=1}^{T} \mathcal{J}(\theta_t, \phi) - \sum_{t=1}^{T} \mathcal{J}(\theta_t, \phi_t).$$

We assume no-regret algorithms for both: $\mathrm{Reg}_G(T) = o(T)$ and $\mathrm{Reg}_D(T) = o(T)$. Let $\bar{\mathcal{J}}_T = \frac{1}{T} \sum_{t=1}^{T} \mathcal{J}(\theta_t, \phi_t)$ denote the average payoff, and let the *game value* be

$$V := \min_{\theta} \max_{\phi} \mathcal{J}(\theta, \phi) = \min_{\theta} \overline{\mathcal{J}}(\theta) = \min_{\theta} \left\{ \mathcal{L}_{\mathrm{BT}}(\theta) + \lambda I_{\theta}(C; R_{\theta}) \right\}.$$

Our next lemma bounds our defined objective $\mathcal{J}$ in terms of the value of the game, with a deviation equal to the average regret of our generator/discriminator algorithms.

**Lemma 2** (No-regret bound for zero-sum play)**.** *Let $\mathcal{J}(\theta, \phi)$ be zero-sum and let a play $(\theta_t, \phi_t)_{t=1}^{T}$ induce*

$$\bar{\mathcal{J}}_T := \frac{1}{T} \sum_{t=1}^{T} \mathcal{J}(\theta_t, \phi_t),$$

$$\mathrm{Reg}_G(T) := \sum_{t=1}^{T} \mathcal{J}(\theta_t, \phi_t) - \min_{\theta} \sum_{t=1}^{T} \mathcal{J}(\theta, \phi_t),$$

$$\mathrm{Reg}_D(T) := \max_{\phi} \sum_{t=1}^{T} \mathcal{J}(\theta_t, \phi) - \sum_{t=1}^{T} \mathcal{J}(\theta_t, \phi_t).$$

*Let $V_{\mathrm{up}} := \min_{\theta} \max_{\phi} \mathcal{J}(\theta, \phi)$ and $V_{\mathrm{low}} := \max_{\phi} \min_{\theta} \mathcal{J}(\theta, \phi)$. Then*

$$V_{\mathrm{low}} - \tfrac{\mathrm{Reg}_D(T)}{T} \;\leq\; \bar{\mathcal{J}}_T \;\leq\; V_{\mathrm{up}} + \tfrac{\mathrm{Reg}_G(T)}{T}. \tag{5}$$

*In particular, if the game has value $V$ (i.e., $V_{\mathrm{up}} = V_{\mathrm{low}} = V$),*

$$\left| \bar{\mathcal{J}}_T - V \right| \;\leq\; \tfrac{\mathrm{Reg}_G(T) + \mathrm{Reg}_D(T)}{T}. \tag{6}$$

*Proof.* We start with the upper bound. By the generator's regret definition,

$$\sum_{t=1}^{T} \mathcal{J}(\theta_t, \phi_t) \;\leq\; \min_{\theta} \sum_{t=1}^{T} \mathcal{J}(\theta, \phi_t) \;+\; \mathrm{Reg}_G(T).$$

Let $\theta^{\star} \in \arg\min_{\theta} \max_{\phi} \mathcal{J}(\theta, \phi)$ (a minimax optimizer). Evaluating the RHS at $\theta^{\star}$ and using $\max_{\phi} \mathcal{J}(\theta^{\star}, \phi) = V_{\mathrm{up}}$ yields

$$\min_{\theta} \sum_{t=1}^{T} \mathcal{J}(\theta, \phi_t) \;\leq\; \sum_{t=1}^{T} \mathcal{J}(\theta^{\star}, \phi_t) \;\leq\; \sum_{t=1}^{T} \max_{\phi} \mathcal{J}(\theta^{\star}, \phi) \;=\; T \, V_{\mathrm{up}}.$$

Combining gives $\sum_{t=1}^{T} \mathcal{J}(\theta_t, \phi_t) \leq T \, V_{\mathrm{up}} + \mathrm{Reg}_G(T)$, hence $\bar{\mathcal{J}}_T \leq V_{\mathrm{up}} + \mathrm{Reg}_G(T)/T$., which completes this part of the inequality.

Next, we demonstrate the lower bound. By the discriminator's regret definition,

$$\sum_{t=1}^{T} \mathcal{J}(\theta_t, \phi_t) \;\geq\; \max_{\phi} \sum_{t=1}^{T} \mathcal{J}(\theta_t, \phi) \;-\; \mathrm{Reg}_D(T).$$

Let $\phi^{\star} \in \arg\max_{\phi} \min_{\theta} \mathcal{J}(\theta, \phi)$ (a maxmin optimizer), so $\min_{\theta} \mathcal{J}(\theta, \phi^{\star}) = V_{\mathrm{low}}$. Then for every $\theta$, $\mathcal{J}(\theta, \phi^{\star}) \geq V_{\mathrm{low}}$. In particular,

$$\max_{\phi} \sum_{t=1}^{T} \mathcal{J}(\theta_t, \phi) \;\geq\; \sum_{t=1}^{T} \mathcal{J}(\theta_t, \phi^{\star}) \;\geq\; \sum_{t=1}^{T} V_{\mathrm{low}} \;=\; T \, V_{\mathrm{low}}.$$

Thus $\sum_{t=1}^{T} \mathcal{J}(\theta_t, \phi_t) \geq T \, V_{\mathrm{low}} - \mathrm{Reg}_D(T)$, i.e., $\bar{\mathcal{J}}_T \geq V_{\mathrm{low}} - \mathrm{Reg}_D(T)/T$.

Combining both sides finishes the proof – in particular, if $V_{\mathrm{up}} = V_{\mathrm{low}} = V$ (minimax theorem of zero-sum games), then

$$V - \frac{\mathrm{Reg}_D(T)}{T} \;\leq\; \bar{\mathcal{J}}_T \;\leq\; V + \frac{\mathrm{Reg}_G(T)}{T},$$

and, since $\max\{a, b\} \leq a + b$ for $a, b \geq 0$, the symmetric bound (6) follows. $\qquad\square$

Another technicality is we require the optimal reward model– the one that satisfies our mutual information constraint while minimizing BT-loss, to lie in our function class. We frame this as the **feasible invariance** condition:

**Feasible invariance.** Let $\mathcal{L}_{\mathrm{BT}}^{\star} = \inf_\theta \mathcal{L}_{\mathrm{BT}}(\theta)$. We say *feasible invariance* holds if there exists $\theta^\dagger$ with

$$\mathcal{L}_{\mathrm{BT}}(\theta^\dagger) = \mathcal{L}_{\mathrm{BT}}^{\star} \quad \text{and} \quad I_{\theta^\dagger}(C; R_{\theta^\dagger}) = 0. \tag{7}$$

In that case, the minimax value satisfies $V = \mathcal{L}_{\mathrm{BT}}^{\star}$ by (4).

With these results, we can then prove our main theorem that in no-regret, our reward model converges to zero mutual-information.

**Proof of Theorem 1 (No Regret Convergence)**

*Proof.* For each $t$, let $V(\theta) = \max_\phi \mathcal{J}(\theta, \phi) = \mathcal{L}_{\mathrm{BT}}(\theta) + \lambda I_\theta(C; R_\theta)$ by Lemma 1. By the discriminator's regret definition,

$$\frac{1}{T} \sum_{t=1}^{T} V(\theta_t) = \frac{1}{T} \sum_{t=1}^{T} \max_\phi \mathcal{J}(\theta_t, \phi) \leq \bar{\mathcal{J}}_T + \frac{\mathrm{Reg}_D(T)}{T}.$$

Feasible invariance implies $V = \mathcal{L}_{\mathrm{BT}}^{\star}$, and Lemma 2 gives $\bar{\mathcal{J}}_T \leq V + \frac{\mathrm{Reg}_G(T)}{T} = \mathcal{L}_{\mathrm{BT}}^{\star} + \frac{\mathrm{Reg}_G(T)}{T}$. Hence

$$\frac{1}{T} \sum_{t=1}^{T} \left[ \mathcal{L}_{\mathrm{BT}}(\theta_t) + \lambda I_{\theta_t}(C; R_{\theta_t}) \right] \leq \mathcal{L}_{\mathrm{BT}}^{\star} + \frac{\mathrm{Reg}_G(T) + \mathrm{Reg}_D(T)}{T}.$$

Since $\mathcal{L}_{\mathrm{BT}}(\theta_t) \geq \mathcal{L}_{\mathrm{BT}}^{\star}$ for all $t$, canceling $\mathcal{L}_{\mathrm{BT}}^{\star}$ yields

$$\lambda \cdot \frac{1}{T} \sum_{t=1}^{T} I_{\theta_t}(C; R_{\theta_t}) \leq \frac{\mathrm{Reg}_G(T) + \mathrm{Reg}_D(T)}{T},$$

which proves the claim. Note that if the average of these terms converges to 0, then we also have that $\inf_t I_{\theta_t} \to 0$, and so we can select the minimum running iterate that is bounded by this average to have a direct convergent subsequence.

$\square$

We view training the discriminator using CELoss on each batch as an approximate "best-response." More formally, we can think of it as an $\epsilon_t$-Nash equilibrium for each round – that is, if $q_{\phi_t}$ is trained to near-optimality per round so that $\max_\phi \mathcal{J}(\theta_t, \phi) - \mathcal{J}(\theta_t, \phi_t) \leq \epsilon_t$ with $\frac{1}{T} \sum_t \epsilon_t \to 0$, then the proof above holds with $\mathrm{Reg}_D(T)$ replaced by $\sum_t \epsilon_t$.

What if exact invariance is infeasible? That is, what if the Bradley-Terry-optimal reward model invariant to category does not lie in our function class? If no $\theta$ attains both $\mathcal{L}_{\mathrm{BT}}^{\star}$ and $I = 0$, then $V > \mathcal{L}_{\mathrm{BT}}^{\star}$ and our theorem instead yields the following bound:

$$\frac{1}{T} \sum_{t=1}^{T} I_{\theta_t}(C; R_{\theta_t}) \leq \frac{V - \mathcal{L}_{\mathrm{BT}}^{\star}}{\lambda} + \frac{\mathrm{Reg}_G(T) + \mathrm{Reg}_D(T)}{\lambda T},$$

where we cannot ignore the $V - \mathcal{L}_{BT}^{*}$ term, which we can think of approximation error-esque term in the learning theory language.

**A.4 Datasets and Preprocessing**

HH-RLHF (Helpful/Harmless): We construct (chosen, rejected) preference pairs and assign each pair a category label of either helpful or harmless. Prompts and responses are concatenated, and sequences are truncated to a maximum of 1,024 tokens.

PKU-SafeRLHF (19 categories): We retain the official harm category labels from the dataset release. Samples with missing category annotations are removed to ensure label integrity.

Deduplication: Exact duplicate (prompt, response) pairs are removed to avoid information leakage and inflated results.

Tokenization and padding: All data is tokenized with padding=longest and truncation=true. Each prompt–response sequence is capped at 1,024 tokens in all reported experiments.

### A.5 Model and Training Details

We use Llama-3.2-1B adapted into a scalar reward model for our RM backbone, with the Bradley-Terry pairwise log-likelihood on (chosen, rejected) pairs as our baseline training objective. We train for a single epoch on a balanced sample of helpful and harmless data from the Anthropic HH-RLHF dataset and evaluate on a held-out set of HH-RLHF dataset as well as RewardBench.

### A.6 Adversary and Fairness Optimization

The fairness constraint uses a lightweight MLP adversary $q_\phi$ that receives summary statistics of rewards, computed separately for each category. For each batch, we calculate the mean, variance, skewness, and kurtosis of the chosen and rejected rewards, grouped by category, to form the adversary's input features.

Our training implementation follows the given alternating update schedule:

1. Compute Bradley–Terry loss $L_{BT} = -\log \sigma(r_{chosen} - r_{rejected})$.

2. Adversary step: update $q_\phi$ by minimizing cross-entropy loss to predict the category from the moment features.

3. Fairness step: update the reward model to maximize adversary uncertainty, i.e., minimize

$$L_{BT} - \lambda_{adv} \cdot CELoss\big(q_\phi(\cdot \mid moments), y\big),$$

Ablation: For ablation studies, we sweep $\lambda_{adv} \in \{0.0, 0.2, 0.4, 0.6, 0.8, 1.0, 1.5, 2.0\}$. The default setting for main experiments is $\lambda_{adv} = 0.2$.

**Post-training Category Predictability.** As a post-training test, we train a fresh discriminator on frozen rewards from the above regularized model, which yields near-chance performance—AUC $0.78 \pm 0.03 \rightarrow 0.53 \pm 0.06$, BA $0.70 \pm 0.02 \rightarrow 0.52 \pm 0.05$ (5-fold; see Appx. A.7)—indicating little recoverable category signal from the fair reward model.

### A.7 Post-hoc Category Predictability Audit

To test whether category information remains after training, we *freeze* the reward model and train a new discriminator $\hat{q}(c \mid r)$ on its scalar outputs (no weights shared with the in-training adversary). We use stratified 5-fold cross-validation and report mean±sd over folds. The discriminator is a 2-layer MLP trained with cross-entropy and early stopping on validation AUC. Chance performance is $0.5$ for both AUC and balanced accuracy (BA).

| Model | AUC | Balanced Acc. |
|---|---|---|
| Baseline RM | $0.78 \pm 0.03$ | $0.70 \pm 0.02$ |
| Fair RM (ours) | $0.53 \pm 0.06$ | $0.52 \pm 0.05$ |

Table 4: Post-hoc predictability from frozen rewards; lower is better (chance $\approx 0.5$).

### A.8 Mutual Information Estimation (Ablation)

We measure the dependence between reward scores and category labels during the $\lambda_{adv}$ sweep. Mutual information (MI) is computed with sklearn.metrics.mutual_info_score between category labels $C \in \{helpful, harmless\}$ and a discretized reward variable, obtained by binning rewards into 50 equal-width bins.

Lower MI indicates that the rewards are more category-independent. As an additional check, we monitor the adversary's balanced accuracy; values close to chance imply minimal category dependence.

## A.9 Full $\lambda_{\text{adv}}$ Sweep

In this section we provide the complete data for our full sweep over adversarial loss parameters.

| $\lambda_{\text{adv}}$ | BT loss | MI |
|---|---|---|
| 0.0 | 2.8712 | 0.2282 |
| 0.2 | 2.2307 | 0.0163 |
| 0.4 | 1.5607 | 0.0088 |
| 0.6 | 1.7104 | 0.0059 |
| 0.8 | 1.1879 | 0.0073 |
| 1.0 | 0.8694 | 0.0141 |
| 1.5 | 0.7432 | 0.0136 |
| 2.0 | 0.8151 | 0.0076 |

Table 5: Complete sweep of $\lambda_{\text{adv}}$ values.

## A.10 Scaling Experiments

To evaluate the scalability of our method, we conducted preliminary experiments on Meta's Llama3-8B-Instruct model on an 8xH100 node. The reward distributions for our Fair-RM variant, shown below, exhibit a more complex, multimodal structure compared to the 1.1B model, which we hypothesize is due to the larger model's capacity to capture finer-grained nuances in the preference data. Despite this, the results confirm that our approach remains effective at scale. There is clear separation between chosen and rejected rewards, indicating preference alignment is maintained. Crucially, the distributions for helpful and harmless categories remain tightly aligned, demonstrating that the fairness constraint successfully generalizes and prevents reward disparities even in larger models. However, both our base model and fair-RM variant achieve around 50% accuracy on a subset of RewardBench after our training, for a variety of reasons but mainly in part due to the small bandwidth we had to only run smaller training runs. Our Fair-RM had on-par performance with the baseline BT model, however, but to achieve SOTA-level eval results on both models, full-scale post-training of RewardBench-competitive models derived from the 8B models is part of our future intended work.

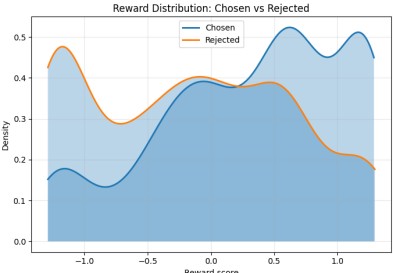

Figure 5: Reward distributions for chosen vs. rejected

Figure 6: Reward distributions for helpful vs. harmless

## B PPO Training Setup

In this section we detail our setup for PPO training of downstream language models using our fair reward models.

**Base Actor.** We initialize all policy variants from `TinyLlama/TinyLlama-1.1B-Chat-v1.0` to enable rapid convergence and reduce compute cost while still maintaining competitive generation quality for our evaluation tasks. Policies are adapted using LoRA with rank $r = 16$ and $\alpha = 32$, targeting the query/key/value and output projection matrices in the attention layers.

**PPO Configuration.** We use HuggingFace TRL's `PPOTrainer` with minibatch size = 64, batch size = 512, and 2 PPO epochs per update. The KL control coefficient is set to $\beta = 0.05$ (adaptive control enabled), targeting the reference model (`TinyLlama/TinyLlama-1.1B-Chat-v1.0`). We set `target_kl`=0.1 to limit divergence from the reference.

**Reward Models.** All reward models are Llama-3.2-1B sequence classifiers trained on preference data with the Bradley–Terry objective. The **Fair** variant applies a mutual information (MI) penalty with $\lambda_{\text{adv}} = 0.2$ between protected-category predictions and reward scores. **Fair + Curiosity** adds an intrinsic curiosity bonus from a Random Network Distillation (RND) module trained online during PPO.

**Curiosity Bonus.** The RND network uses a 2-layer MLP with ReLU activations, hidden size 512. The predictor network is optimized with Adam ($\eta = 1 \times 10^{-4}$) on the cosine similarity loss between target and predictor features. Intrinsic reward is scaled by $\eta_{\text{cur}} = 0.05$ and added to the scalar RM score before PPO optimization.

**Generation Settings.** For PPO rollouts, we generate with temperature $= 0.7$, top-$p = 0.9$, and max length $= 256$ tokens. KL penalties are computed against the reference log-probabilities.

**Training Duration.** Each run is trained for $N = 5{,}000$ PPO steps ($\approx$1.5M tokens processed), which we found sufficient for convergence in both reward and policy loss metrics given the small model size.

## B.1 Parity Gap: Definition and Estimation

In this section we detail a parity gap (effectively mean matching evaluation) for how fair a reward model is, for simplicity across only two categories.

**Definition.** Let $r(x, y)$ denote the scalar reward assigned by a (fixed) safety RM to a prompt–response pair $(x, y)$. We consider two behavior categories $c \in \{\text{Helpful}, \text{Harmless}\}$ and define the *parity gap* as the absolute difference in expected rewards:

$$\text{ParityGap} = \big| \mathbb{E}\big[r(x, y) \mid c = \text{Helpful}\big] - \mathbb{E}\big[r(x, y) \mid c = \text{Harmless}\big] \big|.$$

We define the parity gap as effectively a mean-matching surrogate evaluation – intuitively, a smaller parity gap indicates the RM (and the downstream policy it shapes) treats categories on a comparable reward scale, reducing category-dependent inflation/deflation.

**Estimator.** Given disjoint evaluation sets $\mathcal{D}_{\text{H}}$ and $\mathcal{D}_{\text{A}}$ (Helpful vs. Harmless) with sizes $n_{\text{H}}$ and $n_{\text{A}}$ and rewards $\{r_i^{\text{H}}\}_{i=1}^{n_{\text{H}}}, \{r_j^{\text{A}}\}_{j=1}^{n_{\text{A}}}$, we compute

$$\bar{r}_{\text{H}} = \tfrac{1}{n_{\text{H}}} \sum_{i=1}^{n_{\text{H}}} r_i^{\text{H}}, \qquad \bar{r}_{\text{A}} = \tfrac{1}{n_{\text{A}}} \sum_{j=1}^{n_{\text{A}}} r_j^{\text{A}}, \qquad \widehat{\Delta} = \bar{r}_{\text{H}} - \bar{r}_{\text{A}}, \qquad \widehat{\text{ParityGap}} = |\widehat{\Delta}|.$$

When $n_{\text{H}} \neq n_{\text{A}}$, the above remains unbiased under i.i.d. sampling within each group. In our main runs we use balanced sets ($n_{\text{H}} = n_{\text{A}}$).

**Relative change (vs. a baseline).** When comparing a model $M$ to a baseline $B$, we also report the relative drop:

$$\text{RelDrop}(M; B) = \frac{\widehat{\text{ParityGap}}(M) - \widehat{\text{ParityGap}}(B)}{\widehat{\text{ParityGap}}(B)} \times 100\%.$$

**Practical notes.** (i) We score responses with the same fixed RM across all policies. (ii) Generation settings and seeds are identical across policies (Appendix B).

## B.2 Semantic Diversity Calculation

In this section we detail our metric for diversity of LLM sampling to benchmark our intrinsic reward.

**Prompts and generation.** For diversity evaluation we sample 1,030 LIMA prompts (seed 42) and generate one response per prompt with identical sampling across models. Prompts are drawn from GAIR/lima. Generation parameters: temperature $= 0.9$, top-$p = 0.95$, max_new_tokens$= 100$, max_length$= 512$, batch size $= 8$. All models use the same seed and generation parameters.

**Semantic diversity (primary metric).** Let $f(\cdot)$ be all-mpnet-base-v2 with mean-pooling; embeddings are $\ell_2$-normalized. For the set of responses $\{y_i\}_{i=1}^n$ with embeddings $e_i = f(y_i)$, we report

$$\text{SemDiv} = \tfrac{2}{n(n-1)} \sum_{i<j} \big(1 - \cos(e_i, e_j)\big).$$

Higher is better (more meaning-level variety).

**Statistics.** To compare a fair model against the baseline, we use a paired bootstrap (1,000 resamples; two-sided) over aligned prompt sets, reporting the mean difference, 95% CI, and $p$-value. In the main text, we report semantic-diversity differences: Fair (no curiosity) vs. Baseline: $-0.0054$ ($p<0.001$); Fair + Curiosity vs. Baseline: $-0.0022$ ($p=0.002$).

## B.3 Compute and Runtime

Hardware: For initial experiments of both reward model training and PPO, we used dual A100 clusters, and currently are using a 8xH100 node for results on Llama3-8B.