# OpenReview forum: "GUARD: Guiding Unbiased Alignment through Reward Debiasing"
_NeurIPS.cc/2025/Workshop/Reliable_ML — NeurIPS 2025 - Reliable ML Workshop_

### Official Review · Reviewer_AK9a · 2025-09-20
**category-invariant reward modeling for RLHF**

**Rating:** 7
**Confidence:** 2

**Review:**

*Summary*

The paper targets category-invariant reward modeling for RLHF: the distribution of reward scores should not depend on the prompt/response category. The approach minimizes mutual information between rewards and category labels. Because direct MI minimization is intractable, the authors use an adversarial minimax setup: a classifier tries to predict the category from reward outputs, while the reward model is trained to remove category-specific signal so the classifier fails. To counteract the reduction in generative
diversity, they use a curiosity-driven intrinsic reward. They provide empirical evidence of effectiveness and a theoretical argument that no-regret dynamics drive the system toward the mutual-information target.

*Strengths*

Clear reliability objective, a principled adversarial formulation, and a concise no-regret convergence sketch tying practice to theory.

An improvement compared to earlier work that lack theoretical guarantees of reliability, and may sacrifice response diversity.

*Weaknesses*

All discussions on the theoretical results are only in the appendix; at least the main theorem statements and assumptions should appear in the main text, with proofs deferred to the appendix.

How the curiosity-driven intrinsic reward is incorporated is not formally discussed.

---

### Official Review · Reviewer_DKi9 · 2025-09-21
**Review of GUARD**

**Rating:** 7
**Confidence:** 2

**Review:**

**Summary**

GUARD is a method of debiasing the reward model across different categories by targeting identical reward distributions conditioned on each category. To accomplish this, they construct a classifier to predict the category given the rewards, and frame it as an adversary in a minimax game against the reward model. They also have a "curiosity" variant in which they use small intrinsic rewards to help preserve output diversity.


**Strengths**

This paper is a great fit for the workshop topically. It identifies an elegant and intuitive framing of the problem of reward debiasing as a minimax game, and couples nice theoretical results with strong empirical results.

**Weaknesses**

Because this paper draws on so much existing work (and appears to combine it in a novel way), it's a little difficult to identify what is contributed by this work specifically. I think this can be remedied by just clarifying where this work differs from existing work or specifying exactly what pre-existing techniques it's derivative of.

**Suggestions**

1) **Citation / work positioning clarity:** I found the number of citations in the abstract to be a little overwhelming and it made it difficult to parse. This could just be a stylistic difference, but I think the quantity of citations in the abstract can be reduced (all of them appear in the intro and in related work as well). In the introduction, there's citations for the sentence introducing the paper's key insight: "Our key insight is that..." I found this confusing -- is this key insight not new then? Additionally, in the Related Work section, I think readers would benefit from a clearer separation between this work and previous works. It reads as if the only difference is the incorporation of curiosity-driven rewards on top of what others have already done with adversarial debiasing, but the rest of the paper leads me to believe that is not the case.

3) **Examples of salient categories:** In the abstract and intro, "sensitive categories" / "expert-defined harm categories" are mentioned. I would have found it useful to have a few examples of such sets of categories early on. In fact, the first experiment is not really on "harm categories" -- it compares the "helpful" vs "harmless" categories. However, the second experiment dataset (4.2) is across different stereotype categories. My sense is that the categories all either need to be "positive" or "negative" but it would be useful to say this explicitly somewhere/give examples early on.

5) **Missing runtime info:** This is a very small note, but I believe in Appendix B.3 the runtime is missing.